# Pregnancy Outcomes in Women with Low and Ultra-Low Ejection Fraction: A Retrospective Study in a Tertiary Care Center

**DOI:** 10.3390/jcm14030745

**Published:** 2025-01-24

**Authors:** Bashayer Saeed, Amani ALbalawi, Marwah Bintalib, Amjad Alturki, Edward B. De Vol, Balqees ALzayed, Dania Mohty, Gruschen Veldtman, Maisoon AlMugbel, Nayef Latta, Faisal Joueidi, Wesam Kurdi

**Affiliations:** 1Department of Obstetrics & Gynecology, King Faisal Specialist Hospital & Research Centre, Riyadh 11211, Saudi Arabia; bsaeed83@ymail.com (B.S.); hammad1539@gmail.com (A.A.); mbintalib2@gmail.com (M.B.); am.alturki@hotmail.com (A.A.); mmugbel@kfshrc.edu.sa (M.A.); 2Biostatistics, Epidemiology & Scientific Computing, King Faisal Specialist Hospital & Research Centre, Riyadh 11211, Saudi Arabia; f18568@kfshrc.edu.sa (E.B.D.V.); balzayed@kfshrc.edu.sa (B.A.); 3Cardiovascular Unit, King Faisal Specialist Hospital & Research Centre, Riyadh 11211, Saudi Arabia; mdania@kfshrc.edu.sa (D.M.); gruschen@me.com (G.V.); 4College of Medicine, Alfaisal University, Riyadh 11533, Saudi Arabia; nayeflatta@gmail.com (N.L.); fjoueidi99@gmail.com (F.J.)

**Keywords:** ejection fraction, ultralow, pregnancy, cardiovascular, outcomes, retrospective study

## Abstract

The data about pregnancy while having a low ejection fraction are scarce, since pregnancy is not recommended for women with an ejection fraction of less than 30%. There is an increased risk of obstetrical complications and adverse maternal-fetal outcomes. Pregnancy is a rough journey for this group of patients. However, a successful pregnancy can be achieved when cardiac complications are managed during pregnancy. The early recognition of women at risk of cardiovascular events and early referral can optimize the maternal and neonatal outcomes with close collaboration between the maternal-fetal medicine specialist and the cardiologist. The study’s aim was to assess the experience of our tertiary center with regard to the adverse maternal outcome for women with an ejection fraction ≤ 30% compared to those with an EF > 30% in our tertiary center. The fetal and obstetric outcome for pregnancies with an EF ≤ 30% was compared to that for pregnancies with an EF > 30%. **Methodology**: After receiving the approval of the local Ethical Board Review, a retrospective study was conducted at King Faisal Specialist Hospital and Research Center (KFSHRC) in the city of Riyadh, Kingdom of Saudi Arabia. Our study population included women with cardiomyopathy (acquired or congenital) who were followed up or delivered in KFSHRC from the period of January 2004 till March 2020. Cases were identified by reviewing the database from the Cardiac Center Echocardiograph and maternal fetal medicine unit. The data on the maternal and fetal outcome were gathered from the hospital medical records. An adverse maternal outcome included: death, hospitalization due to decompensated heart failure, and worsening cardiovascular status during pregnancy. Adverse fetal outcomes included: miscarriages, termination of pregnancy, FGR, and placental insufficiency. Obstetrics complications included: complications related to the mode of delivery, antepartum hemorrhage, postpartum hemorrhage, or preeclampsia. **Results**: Our study included 44 subjects, examining the differences between those with an ejection fraction greater than 30 (n = 21 subjects) and those with an ejection fraction less than or equal to 30 (n = 23) with respect to demographics, co-morbidities, and outcomes (maternal, pregnancy, fetal, ultrasound, and baby). There was no evidence of any differences in the demographics. From among the co-morbidities, there was a statistically higher rate of dilated cardiomyopathy and lower rate of rheumatic heart disease in those with a lower ejection fraction. Also, women with a lower ejection fraction tended to deliver through a means other than simple vaginal delivery. There was a significant association (*p* = 0.0296) indicating that individuals with a lower ejection fraction tended to have a lower gestational age at delivery. The information on whether the pregnancy resulted in a live birth was available for all but one of the mothers. Across all the mothers, 32 (74%) resulted in a live birth and 11 did not. The percentage of pregnancies resulting in a live birth in the group for which the ejection fraction was greater than 30 was 90% and that in the group for which the ejection fraction was less than or equal to 30 was 59% (*p* = 0.0339). From among the ultrasound and baby outcomes, only the rate of the babies being discharged alive differed statistically between the two ejection fraction groups, with those mothers having a lower ejection fraction experiencing fewer babies being discharged alive (*p* = 0.0310). **Conclusions**: In conclusion, women with a low ejection fraction are at an increased risk of maternal-fetal complications. In our study, the lower the EF (≤30) the worse were the fetal and neonatal outcomes; however, in terms of the maternal outcomes, it was the same whether the EF was low or ultra-low. Yet, these groups of patients need to be counseled about the facts of poor obstetrical outcomes with an emphasis on preconceptual counseling.

## 1. Introduction

The number of women with pre-existing cardiovascular disease or developing cardiac problems during pregnancy are increasing worldwide [1]. For example among the 4 million pregnancies in the US each year, cardiac diseases affect around 1–4% with a mortality rate reaching 25–30% of all maternal deaths. The leading cause of maternal mortality is due to cardiomyopathy, accounting for around 50% of cases [2,3].

The rising rates of advanced maternal age, preeclampsia, multiple gestation, obesity, diabetes mellites, and hypertension in pregnancy have been attributed as risk factors for cardiomyopathies [3]. Of course, these risk factors are not unique to the US. The data from a national survey conducted in the Kingdom of Saudi Arabia (KSA) between 1996 and 2011 showed that over a period of ten years, the prevalence of obesity increased in Saudi women from 23.6% to 44.0%; self-reported physical inactivity worsened from 84.7% to 98.1%; the prevalence of smoking in women increased from 0.9% to 7.6%. The prevalence of metabolic syndrome was significantly greater in women than men (42.0% versus 37.2%; *p* < 0.01). All of this indicates that Saudi women are potentially at a greater risk than a decade ago of developing cardiovascular diseases and diabetes mellitus, with a notable increase in obesity [4].

A normal pregnancy imposes dramatic physiological changes upon the cardiovascular system, such as an increase in plasma volume by 50%, an increase in the resting pulse by 17%, and an increase in cardiac output by 50% [5]. The cardiovascular system undergoes specific adaptations to meet the increased demands of the mother and fetus during pregnancy. These significant hemodynamic changes that accompany pregnancy make the diagnosis of certain forms of cardiovascular disease difficult [6].

Women with cardiac disease in pregnancy may tolerate these physiological changes well; however, they are considered to have a high-risk pregnancy due to the maternal and fetal complications that might occur. If obstetric complications do occur, this can have a significant impact on the outcomes of pregnancy. For example, preeclampsia increases the risk of cardiac decompensation and death, and postpartum hemorrhage can lead to hypovolemic shock, which is often poorly tolerated [6]. Pregnancy is considered a rough journey with the risk of heart failure, arrythmias, thromboembolic events, and even maternal death [7]. Pregnant women with advanced heart disease, especially those with low cardiac output or severe hypoxia, experience a greatly increased incidence of fetal loss in the first trimester, intrauterine fetal death, intrauterine growth restriction, and prematurity [6].

In a retrospective study of the incidence of fetal growth restriction (FGR) in cardiac pregnant women, the rate was 9.15% [8]. Fetal growth restriction is of rising concern not just for its risks and complications intrauterine and early neonatal period but also for the future health burden that it holds for the future offspring. The long-term consequences due to changes in the fetal nutritional environment are associated with an increased risk of developing metabolic syndrome and cardiovascular disease, systolic hypertension, obesity, insulin resistance, and type II diabetes in adulthood [9].

Although the journey is rough, a successful pregnancy can be achieved when cardiac complications are managed during pregnancy. The early recognition of women at risk of cardiovascular events and early referral can optimize the maternal and neonatal outcomes with close collaboration between the maternal-fetal medicine specialist and the cardiologist [5].

In our study, we aimed to look at:-The adverse maternal outcomes for women with an ejection fraction ≤30% compared to those with an EF > 30% in our tertiary center.-The fetal and obstetric outcome for pregnancies with an EF ≤ 30% compared to pregnancies with an EF > 30%.

## 2. Methodology

After receiving the approval of the local Ethical Board Review, a retrospective study was conducted at King Faisal Specialist Hospital and Research Center (KFSHRC) in the city of Riyadh, Kingdom of Saudi Arabia.

Our study population included women with cardiomyopathy (acquired or congenital) who were followed up or delivered in KFSHRC from the period of January 2004 till March 2020. Cases were identified by reviewing the database from the Cardiac Center Echocardiograph and maternal fetal medicine unit. The data on the maternal and fetal outcome were gathered from the hospital medical records.

An adverse maternal outcome included: death, hospitalization due to decompensated heart failure, and worsening cardiovascular status during pregnancy. Adverse fetal outcomes included: miscarriages, termination of pregnancy, FGR, and placental insufficiency. Obstetric complications included: complications related to the mode of delivery, antepartum hemorrhage, postpartum hemorrhage, or preeclampsia.

## 3. Results

Table 1 illustrates the differences between those with an ejection fraction (EF) greater than 30 (n = 21 subjects) and those with an ejection fraction less than or equal to 30 (n = 23) with respect to the demographics, co-morbidities, and outcomes (maternal, pregnancy, fetal, ultrasound, and baby). There was no evidence of differences in the demographics. From among the co-morbidities, there was a statistically higher rate of dilated cardiomyopathy and lower rate of rheumatic heart disease in those with a lower ejection fraction. In 25% of the study subjects, the termination of pregnancy was carried out to avoid adverse maternal outcomes, 39% in a lower EF group in contrast to 10% in a higher EF group (*p* = 0.0365). Also, the women with a lower ejection fraction tended to deliver through means other than simple vaginal delivery (*p* = 0.0092). The incidence of spontaneous vaginal births (50%) and emergency caesarean sections (28%) were higher in the women with a higher EF. On the other hand, the miscarriage rate (40%) and elective caesarean delivery (35%) were observed in the ≤30% ejection fraction group. There was a significant association (*p* = 0.0296) indicating that individuals with a lower ejection fraction tended to have a lower gestational age at delivery.

The information on whether the pregnancy resulted in a live birth was available for all but one of the mothers. Across all the mothers, 32 (74%) resulted in a live birth and 11 did not. The percentage of pregnancies resulting in a live birth in the group for which the ejection fraction was greater than 30 was 90% and that in the group for which the ejection fraction was less than or equal to 30 was 59% (*p* = 0.0339).

From among the ultrasound and baby outcomes, only the rate of the baby being discharged alive differed statistically between the two ejection fraction groups, with those mothers having a lower ejection fraction experiencing fewer babies being discharged alive (*p* = 0.0310).

In addition, we wanted to investigate the effect of the time to an event, i.e., the number of days of conception that were tolerated till the patient required hospitalization or ended with death. From the figures above, 12 out of n = 44 subjects required hospitalization due to heart failure; 75% of the subjects did not need hospitalization for cardiovascular decompensation vs 25% who needed hospitalization due to heart failure (Figure 1, Table 1 maternal outcome).

Figure 2 describes the number of days of pregnancy till mortality as an event. From the above, we had five mortalities that occurred between the gestational age of 100 and 250 days (18–36 weeks GA). Later, we performed a comparison between subjects with a low EF vs an ultra-low EF in terms of the difference in hospitalization or death. There was no evidence of a difference in the time to hospitalization or death between the groups (Figure 3 and Figure 4).

## 4. Discussion

Most of the research work on cardiac disease in pregnancy has been performed to find the different causes and their demographic associations. We compared women with cardiac disease in two subgroups with different ventricular ejection fractions: having either a >30% ejection fraction (n = 21 subjects) or ≤30% EF (n = 23 subjects). The study demonstrates a correlation between a lower ejection fraction and a higher chance of having adverse maternal, fetal, pregnancy, and neonatal outcomes. In our study population, dilated cardiomyopathy was the commonest finding in the low ejection fraction group in contrast to rheumatic heart disease which was the rarest. Those with a lower EF had a higher termination of pregnancy rate and tended to deliver through means other than vaginal delivery. Also, this group had a higher rate of neonatal prematurity, lower rate of live births, and tended to have less babies being discharged alive from the hospital. The results met most of our expectations and supported our hypotheses (https://www.scribbr.com/methodology/hypothesis/ (accessed on 25 July, 2024)). We could see the strength of the association of our findings within the previous research and theory meeting the Bradford Hill criteria of temporality and consistency [10]. Hence, such novel finings can be extrapolated for other patient papulations.

In spite of the higher rate of prematurity mean of 29.9 ± 9.8 weeks, the rate of live births was 74%, and the rate of newborns discharged alive was 72%. The reason may be our excellent NICU facilities, patients being managed under multidisciplinary care, and the careful monitoring of the maternal-fetal status in cardiac patients with a lower EF. Also, in most of the cases, the careful monitoring of women in labor with heart disease via continuous cardiotocography in order to decrease the rate of perinatal morbidity was essential. Similar to our findings, the research by Monteiro AV [11] demonstrated a deterioration of the maternal cardiac function and a higher rate of miscarriage, a small for gestational age newborns, and APGAR score < 7 in the group with an impaired left ventricular fraction.

In line with the research work of Kampman MA, [12] we also noted significant changes in the maternal cardiac condition when going through the second half of pregnancy with birth requiring hospital admission. This finding highlighted the importance of echocardiographic assessment during pregnancy and serial follow-up with multidisciplinary care provided [12].

The management of cardiac disease in pregnancy also depends upon the type, duration, severity, and extent of the structural damage to the heart, functional status of the cardiovascular system of women, as well as the hemodynamic status of patients. Stratification models, such as the Canadian Cardiac Disease in Pregnancy risk index (CARPREG II) (a comprehensive scoring system that incorporates general cardiac factors, specific cardiac lesions, and the process of care factors), the Zwangerschap bij Aangeboren HARtAfwijkingen (ZAHARA) (a weighted risk score for congenital heart disease patients), and the modified World Health Organization (WHO) classification of maternal cardiovascular risk are various risk assessment tools that can aid clinicians in the management of women during pregnancy or their postpartum period [2]. Although most of our patients were known cardiac patients on different therapies and being followed up in a tertiary care center, they still developed problems during their antenatal, intrapartum, and postpartum periods. The incidence of fetal complications in all the trimesters of pregnancy, labor, perinatal, early neonatal, and late neonatal periods remained high compared to women with an LVEF > 30%.

Maternal cardiac diseases remain a threat to the safety of the mother and fetus, a threat that can extend to impact the future health of both. The Registry Of Pregnancy And Cardiac disease (ROPAC) under the umbrella of the European Society of Cardiology (ESC) reported an overall 0.6% maternal mortality from cardiac disease, with the most common cause being pulmonary arterial hypertension [13]. In our study, five maternal mortalities occurred due to an underlying cardiac disease between the gestational age of 100 and 250 days. The incidence of maternal mortality from cardiac diseases in KSA remains unknown due to the rarity of published data. ALAtawi F [14] published a study that described the rationale and study design for the Registry Of Saudi Heart Disease And Pregnancy (ROSHDAP), which will be a five-phase program plan including initiation, development, prelude, nationwide, and conclusion phases. Cardiac diseases are a group of many different structural and functional disturbances. We need to further explore the effects of pregnancy on heart disease prognosis.

In this study, none of our cases were contacted as we retrieved the information from a database, so we cannot provide any information on the long-term health of these women. Ideally, women with severe cardiac disease should be advised against embarking on a new pregnancy unless their condition has been well controlled via the input of a multidisciplinary team. Counseling women to avoid pregnancy might be a challenge due to cultural and religious issues and due to their desire for larger families. The fact that most patients in this region associate fertility with femininity may also be a cause for refusal [15].

## 5. Conclusions

In conclusion, women with a low ejection fraction are at an increased risk of maternal-fetal complications. In our study, the lower the EF (≤30), the worse were the fetal and neonatal outcomes; however, in terms of the maternal outcomes, it was the same whether the EF was low or ultra-low. Yet, these groups of patients need to be counseled about the facts of poor obstetrical outcomes with an emphasis on preconceptual counseling.

## 6. Future Recommendations

Preconceptual counseling should aim to decrease the absolute rate of unplanned and planned pregnancies in this group of women by using a more cautious approach in educating women and their partners about the seriousness and implications of life-threatening cardiac diseases.Awareness campaigns, improvements in antenatal care and the counseling of patients preconceptually are needed.

## Figures and Tables

**Figure 1 jcm-14-00745-f001:**
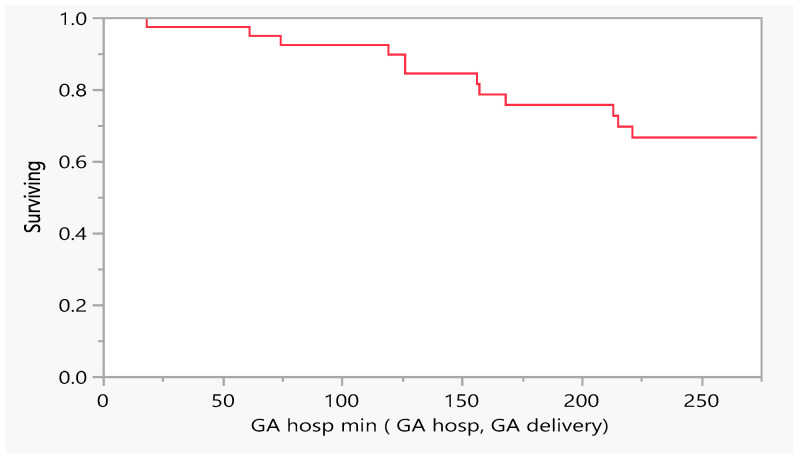
Product-Limit Survival Fit. Survival Plot. Time to event: GA/hospitalization min (GA hosp, GA delivery), GA= gestational age.

**Figure 2 jcm-14-00745-f002:**
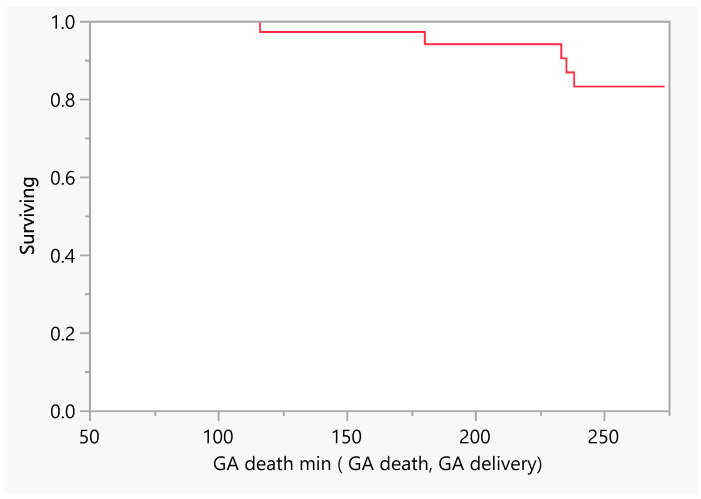
Product–Limit Survival Fit. Survival Plot. Time to event: GA death min (GA death, GA delivery). GA = gestational age.

**Figure 3 jcm-14-00745-f003:**
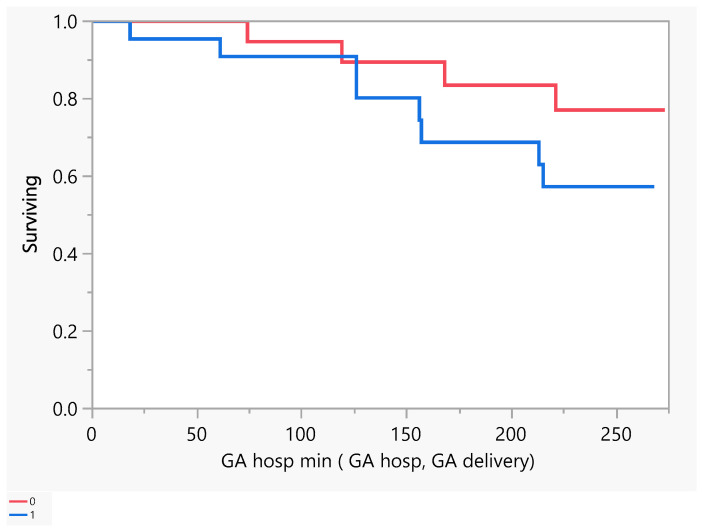
Product-Limit Survival Fit. Survival Plot. Time to event: GA hosp min (GA hosp, GA delivery). Censored by Hospitalized for Heart Failure. Red = EF > 30, blue = EF ≤ 30.

**Figure 4 jcm-14-00745-f004:**
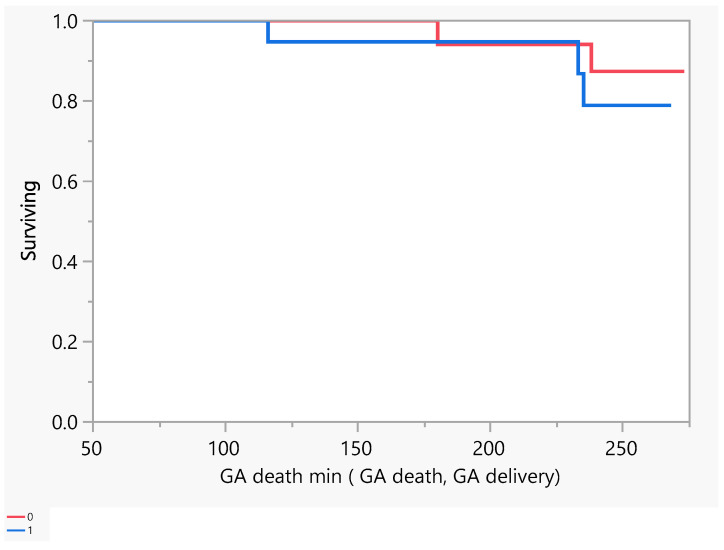
Product-Limit Survival Fit. Survival Plot. Time to event: GA death min (GA death, GA delivery). Red = EF > 30, blue = EF ≤ 30.

**Table 1 jcm-14-00745-t001:** Maternal outcome.

Variable		All Pregnancies (n = 44) Mean ± SD or Frequency (%)	Pregnancies EF > 30 (n = 21) Mean ± SD or Frequency (%)	Pregnancies EF ≤ 30 (n = 23) Mean ± SD or Frequency (%)	*p* Value
Demographics					
Age (years)		34.1 ± 6.5	33.3 ± 6.9	34.8 ± 6.1	0.466
Age > 35		20 (45%)	8 (38%)	12 (52%)	0.381
Gravida n = 41		4.4 ±3.2	4.5 ± 2.8 (n = 20)	4.3 ± 3.5 (n = 21)	0.867
Para n = 41		2.4 ± 2.6	2.5 ± 2.1 (n = 20)	2.4 ± 3.0 (n = 21)	0.979
BMI (kg/m)		29.2 ± 6.2	29.6 ± 5.5	28.8 ± 6.8	0.630
BMI	Underweight < 18.5	1 (2%)	0	1 (4%)	0.356
	Normal = (18.5–24.9)	9 (20%)	3 (14%)	6 (26%)	
	Over = (25–29.9)	16 (36%)	10 (48%)	6 (26%)	
	Obese > 30	18 (41%)	8 (38%)	10 (43%)	
BMI obese n = 18	Class 1 (30–34.9)	11 (61%)	5 (63%)	6 (60%)	1.000
	Class 2 (35–39.9)	5 (28%)	2 (25%)	3 (30%)	
	Class 3 >40	2 (11%)	1 (13%)	1 (10%)	
Abortions n = 41		1 ± 1.5	1.0 ± 1.5 (n = 20)	1.0 ± 1.5 (n = 21)	0.756
LV EF%		30.9 ± 8.8	38.4 ± 3.8	24.0 ± 6	<0.0001 *
Comorbid illness					
Pulmonary HTN		19 (43%)	8 (38%)	11 (48%)	0.556
Diabetes		9 (20%)	3 (14%)	6 (26%)	0.461
Hypertension		8 (18%)	4 (19%)	4 (17%)	1.000
Dyslipidemia		3 (7%)	2 (10%)	1 (4%)	0.598
Chronic Kidney Disease		2 (5%)	2 (10%)	0	0.222
Lupus nephritis		6 (14%)	3 (14%)	3 (13%)	1.000
Previous history of deep venous thrombosis		7 (16%)	3 (14%)	4 (17%)	1.000
Hypothyroidism		3 (7%)	0	3 (13%)	0.234
Systemic lupus erythematosus		7 (16%)	3 (14%)	4 (17%)	1.000
Antiphospholipid syndrome		6 (14%)	3 (14%)	3 (13%)	1.000
Breast cancer		1 (2%)	0	1 (4%)	1.000
Leukemia		2 (5%)	0	2 (9%)	0.489
Wilms tumor		1 (2%)	0	1 (4%)	1.000
Immune Thrombocytopenic Purpura		3 (7%)	3 (14%)	0	0.1004
Smoking		0	0	0	NA
New York Heart Association Functional Classification (NYHA) Score	1	22 (50%)	12 (57%)	10 (43%)	0.7133
	2	10 (23%)	5 (24%)	5 (22%)	
	3	8 (18%)	3 (14%)	5 (22%)	
	4	4 (9%)	1 (5%)	3 (13%)	
Dilated cardiomyopathy		23 (52%)	6 (29%)	17 (74%)	0.0059 *
Rheumatic heart disease		14 (32%)	12 (57%)	2 (9%)	0.0009 *
Tricuspid stenosis		0	0	0	NA
Aortic stenosis		2 (5%)	1 (5%)	1 (4%)	1.000
Pulmonary stenosis		0	0	0	NA
Mitral regurgitation		23 (52%)	10 (48%)	13 (57%)	0.763
Tricuspid regurgitation		18 (41%)	8 (38%)	10 (43%)	0.7667
Aortic regurgitation		7 (16%)	5 (24%)	2 (9%)	0.231
Pulmonary regurgitation		2 (5%)	1 (5%)	1 (4%)	1
Postpartum induced cardiomyopathy		0	0	0	NA
Chemotherapy-induced cardiomyopathy		2 (5%)	0	2 (9%)	0.489
Tetralogy of Fallot (TOF)		3 (7%)	0	3 (13%)	0.234
Ischemic heart disease		4 (9%)	2 (10%)	2 (9%)	1.000
Mitral stenosis		5 (11%)	4 (19%)	1 (4%)	0.176
Heterotopic atrial tachycardia		2 (5%)	2 (10%)	0	0.222
Atrioventricular block		5 (11%)	4 (19%)	1 (4%)	0.176
Maternal outcome					
Patient died		5 (11%)	2 (10%)	3 (13%)	1.000
Hospitalized for heart failure during pregnancy		12 (27%)	4 (19%)	8 (35%)	0.318
Atrial fibrillation		7 (16%)	4 (19%)	3 (13%)	0.692
Atrial flutter		2 (5%)	2 (10%)	0	0.222
Gestational diabetes		1 (2%)	0	1 (4%)	1.000
Pregnancy outcome					
Termination		11 (25%)	2 (10%)	9 (39%)	0.0365 *
Placental insufficiency n = 42		4 (10%)	2 (10%)	2 (9%)	1.000
Premature ruptured membrane n = 40		4 (10%)	1 (5%)	3 (14%)	0.607
Placental abruption n = 42		1 (2%)	0	1 (5%)	1.000
Type of Delivery N = 38	Vaginal Delivery	11 (29%)	9 (50%)	2 (10%)	0.0092 *
	Emergency C-Section	8 (21%)	5 (28%)	3 (15%)	
	Elective C-section	9 (24%)	2 (11%)	7 (35%)	
	Abortion	10 (26%)	2 (11%)	8 (40%)	
Fetal outcome					
Gestational age at delivery n = 37		29.9 ± 9.8	33.5 ± 6.9 (n = 17)	26.8 ± 10.9 (n = 20)	0.0296 *
Live birth n = 43		32 (74%)	19 (90%)	13 (59%)	0.0339 *
Still birth n = 40		1 (3%)	0	1 (5%)	1.000
Miscarriage n = 43		10 (23%)	2 (10%)	8 (36%)	0.068
Prematurity n = 38		13 (34%)	6 (33%)	7 (35%)	1.000
Intrauterine growth restriction n = 38		3 (8%)	2 (11%)	1 (5%)	0.594
Oligohydramnios n = 38		1 (3%)	0	1 (5%)	1.000
Fetal distress n = 38		3 (8%)	2 (11%)	1 (5%)	0.594
Ultrasound findings					
Umbilical artery Doppler PI n = 21		1.0 ± 0.2	1.1 ± 0.2 (n = 12)	1.0 ± 0.2 (n = 9)	0.647
Fetal birth weight US (g) n = 25		1964.04 ± 806.958	1864.071 ± 713.696 (n = 14)	2091.273 ± 932.131 (n = 11)	0.511
Middle cerebral artery (MCA) PI n = 7	Normal	6 (86%)	4 (100%)	2 (67%)	0.428
	low	1 (14%)	0	1 (33%)	
Biophysical Profile (BPP) n = 25		8 ± 0 (n = 25)	8 ± 0 (n = 14)	8 ± 0 (n = 11)	NA
Amniotic fluid index (cm) n = 22		13.6 ± 4.5	13.74 ± 3.05 (n = 12)	13.38 ± 5.99 (n = 10)	0.866
Baby outcome					
Gender n = 26	male	14 (54%)	8 (50%)	6 (60%)	0.701
	female	12 (46%)	8 (50%)	4 (40%)	
Fetal birth weight on delivery (g) n = 23		2302.4 ± 745.4	2209.6 ± 732.2 (n = 13)	2423 ± 784.1 (n = 10)	0.513
Discharged alive n = 39		28 (72%)	17 (89%)	11 (55%)	0.0310 *
AS 1 min n = 23		7.04 ± 1.99	7.15 ± 1.99 (n = 13)	6.9 ± 2.08 (10)	0.770
AS at 5 min n = 23		8.8 ± 0.9	8.7 ± 0.9 (n = 13)	8.9 ± 0.7 (n = 10)	0.560
Cord PH Arterial n = 19		7.241 ± 0.097	7.217 ± 0.092 (n = 9)	7.26 ± 0.100 (n = 10)	0.310
Cord PH Venous n = 20		7.28 ± 0.08	7.27 ± 0.10 (n = 10)	7.29 ± 0.07 (n = 10)	0.459

HTN: Hypertension, * *p* < 0.05, PI: Pulsatility Index; AS: APGAR score.

## Data Availability

The data presented in this study are available on request from the corresponding author due to restrictive access to hospital information systems.

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
