# Peer review of "Pregnancy Outcomes in Women with Low and Ultra-Low Ejection Fraction: A Retrospective Study in a Tertiary Care Center"

_jcm, 2025, doi:10.3390/jcm14030745_

Round 1

Reviewer 1 Report

Comments and Suggestions for Authors

The authors, in their paper titled "Pregnancy Outcomes in Women with Low and Ultra-low Ejection Fraction: A Retrospective Study in a Tertiary Care Center," provide an insightful overview of pregnancy outcomes in women with heart failure. Their findings indicate that both patients with low and ultra-low ejection fractions experienced worse outcomes for both mother and newborn.

I have a few suggestions to enhance the overall quality of the paper:

  1. Including a central figure summarizing the key findings would add significant value to the paper.

  2. In lines 133-134, you state, "There was a suggestion (p = 0.0296) that those with a lower ejection fraction had a lower gestational age at delivery." However, the p-value actually reaches statistical significance.

  3. The introduction should expand on the different presentations and sex-related differences in the etiologies of heart failure, as this is important for understanding the rest of the paper. Please refer to the following comprehensive review on the topic: PMID: 37504533.

Comments on the Quality of English Language

An English language revision should be performed throughout the manuscript

Author Response

Thank you for your careful review, please find the following reply to the comments:

Comment: In lines 133-134, you state, "There was a suggestion (p = 0.0296) that those with a lower ejection fraction had a lower gestational age at delivery." However, the p-value actually reaches statistical significance.

Answer:  corrections done

Comment: The introduction should expand on the different presentations and sex-related differences in the etiologist of heart failure Answer: the study was retrospective from medical records hence we did not have all data on patient presentation plus all our patient were females.

Reviewer 2 Report

Comments and Suggestions for Authors

In the retrospective study, the authors included pregnant women with cardiomyopathy (acquired or congenital heart) with LVEF<30% and >30% to compare the clinical outcomes. They evaluated death, hospitalization due to decompensated heart failure, worsening cardiovascular status during pregnancy. The authors found that pregnant women with low ejection fraction are at increased risk of maternal-fetal complications. However, I would like to make some comments.

1. The authors should look through the text of the paper and correct LVEF<30% / >30% as LVEF<=30%/>30%.

2. Please, add flow chart of the study, clearly describe the design ad indicate the criteria of inclusion / exclusion.

3. Please, add thorough explanation of the concomitant medications and methods of initropic support when relevant.

4. According to table 1 the women with peripartum CMP were not included, so please, give clear report regarding etiology of the HFrEF

5. SEction Discussion. Please, extend the comment regardog novelty opf the findings and the possibility to extrapolate these through other patients' populations.

Comments on the Quality of English Language

The mauscript contains many typos / errors and requires to be corrected by native speaker

Author Response

Thank you for your careful review, please find the following reply to the comments:

Comment: The authors should look through the text of the paper and correct LVEF<30% / >30% as LVEF<=30%/>30%.

Reply: Revised, thanks

Comment: Please, add thorough explanation of the concomitant medications and methods of initropic support when relevant.

Reply: We did not include the medication/inotropes during data collection.

Comment: According to table 1 the women with peripartum CMP were not included, so please, give clear report regarding etiology of the HFrEF

Reply: Peripartum CMP cases were not included in the study as this disease entity is known to have dismal outcomes to both mother and fetus which might bias the results.

Comment:  SEction Discussion. Please, extend the comment regardog novelty opf the findings and the possibility to extrapolate these through other patients' populations.

Reply: comment added

Round 2

Reviewer 1 Report

Comments and Suggestions for Authors

All my observations have been answered.

Comments on the Quality of English Language

Fine

Reviewer 2 Report

Comments and Suggestions for Authors

The authors resubmitted the revised version of the manuscript along with a clear respond to reviewers. The article got much better after the authors focused on the clinical findings and emphasised to the readers the uniqueness of the findings. I am satisfied with the revision and have no serious concerns about the paper in its revised version. The paper